# Effects of digital chatbot on gender attitudes and exposure to intimate partner violence among young women in South Africa

**Alexandra De Filippo[1], Paloma Bellatin[1], Neville Tietz[2], Eli Grant[3], Alexander Whitefield[4,5], Puseletso Nkopane[2], Camilla Devereux[4], Kaitlyn Crawford[2], Benjamin Vermeulen[2], Abigail M. Hatcher [6,7] ***

**1** Behavioral Insights Team, New York, New York, United States of America, **2** Praekelt.org, Johannesburg, South Africa, **3** Independent consultant, Connecticut, United States of America, **4** Behavioural Insights Team, London, United Kingdom, **5** Department of Economics, Duke University, Durham, North Carolina, United States of America, **6** Department of Health Behavior, Gillings School of Global Public Health, University of North Carolina, Chapel Hill, North Carolina, United States of America, **7** School of Public Health, Faculty of Health Sciences, University of the Witwatersrand, Johannesburg, South Africa

* abbeymae@email.unc.edu

**Data Availability Statement:** Data is fully available without restriction at our trial registry: https://osf.io/jcu4n.

## Abstract

### Background

South Africa has among the highest rates of intimate partner violence (IPV) globally, with young women at heightened risk due to inequitable gender roles, limited relationship skills, and inadequate social support. Despite an urgent need for violence prevention in low- and middle-income settings, most efficacious approaches are time-intensive and costly to deliver. Digital, interactive chatbots may help young women navigate safer relationships and develop healthier gender beliefs and skills

### Methods

Young women (18–24 years old) across South Africa were recruited via *Facebook* for participation in an individually randomised controlled trial (*n* = 19,643) during the period of June 2021-September 2021. Users were randomly allocated, using a pipeline algorithm, to one of four trial arms: *Pure Control (PC)* had no user engagement outside of study measures; *Attention Treatment (T0)* provided didactic information about sexual health through a text-based chatbot; *Gamified Treatment (T1)* was a behaviourally-informed gamified text-based chatbot; *Narrative Treatment (T2)* was a behaviourally-informed drama delivered through pre-recorded voice notes. All chatbots were delivered in *WhatsApp*, through which users were invited to complete brief "quizzes" comprising adapted versions of validated scales. Primary outcomes were short-form adaptations of scales for gender attitudes (*Gender Relations Scale*) and past-month IPV (*WHO Multi-country Study Instrument*). Secondary outcomes were identification of unhealthy relationship behaviours (*Intimate Partner Violence Attitudes Scale*) and brief screener for depressive symptoms (*Patient Health Questionnaire*). A direct chat link to a trained counsellor was a safety measure (accessed by 4.5% of

**Funding:** This work was supported by Wellspring Philanthropic Fund to BV and National Institute of Mental Health (K01MH121185 to AMH). The funders had no role in study design, data collection and analysis, decision to publish, or preparation of the manuscript.

**Competing interests:** The authors have declared that no competing interests exist.

the sample). We estimated treatment effects using ordinary least squares and heteroskedasticity robust standard errors

## Findings

The trial retained 11,630 (59.2%) to the primary endpoint of gender attitudes. Compared to control, all treatments led to moderate and significant changes in attitudes towards greater gender equity (Cohen's D = 0.10, 0.29, 0.20 for T0, T1, and T2, respectively). The gamified chatbot (T1) had modest but significant effects on IPV: 56% of young women reported past-month IPV, compared to 62% among those without treatment (marginal effects = -0.07, 95%CI = -0.09to-0.05). The narrative treatment (T2) had no effect on IPV exposure. T1 increased identification of unhealthy relationship behaviours at a moderate and significant level (Cohen's D = 0.25). Neither T1 nor T2 had a measurable effect on depressive symptoms as measured by the brief screener. **Interpretation:** A behaviourally-informed, gamified chatbot increased gender equitable attitudes and was protective for IPV exposure among young women in South Africa. These effects, while modest in magnitude, could represent a meaningful impact given potential to scale the low-cost intervention.

## Author summary

Intimate partner violence (IPV) affects one in three women worldwide and harms health and well-being. Yet, there are few low-cost ways proven to prevent IPV. We tested if digital, interactive chatbots could help young women across South Africa stay safer by changing their attitudes and skills. We assigned similar young women to one of two chatbots. The first was gamified (with emojis and small tasks through *Whatsapp* chat). The second was narrative (with a story pre-recorded using *Whatsapp* voice notes). Their responses were compared to young women randomly assigned to a control group that received information-only or no chatbot at all. Both gamified and narrative chatbots improved young women's attitudes and increased skills in identifying unhealthy relationship. The gamified chatbot only reduced IPV after three months. This is one of the first digital approaches to decrease IPV exposure.

## Introduction

Exposure to intimate partner violence [IPV] among South African women is higher than most settings globally. One in five women report lifetime IPV exposure [1], and rates of femicide are six times the global average [2]. Younger women report higher rates of IPV compared to older peers [3], and in one study 53% of young women aged 14–19 years reported physical and/or sexual IPV [4]. Subtler forms of unhealthy relationship behaviours are also prominent for young women, including age-disparate relationships, early unintended pregnancies, and relationships characterised by controlling partners [1,5,6].

Extant cross-sectional and qualitative literature points to factors associated with IPV exposure among young women. Inequitable gender attitudes [7,8], inadequate social support [9], and limited knowledge of harmful relationship behaviours [10], are associated with higher IPV exposure. Building skills around autonomy and agency in relationships is tied to lower IPV exposure among young women [11,12].

Experimental data from sub-Saharan Africa include *SASA*! in Uganda in which critical discussions around gender reduced IPV exposure among women [13], and *IMAGE* South Africa and Tanzania, where group-based gender workshops combined with microfinance reduced adult women's IPV [14,15]. Yet, of 36 African trials reviewed in 2022, only one-third reduced IPV among women [16]. Similarly, few rigorously tested approaches have been shown to reduce men's perpetration of IPV in African trials. Group-based gender training with younger men may decrease reported IPV perpetration [17], particularly if combined with economic livelihoods [18]. However, other trials have had null findings for men's IPV perpetration [13,19,20].

Mobile devices offer a promising channel to deliver targeted edutainment at scale through mHealth, a strategy shown to effectively encourage health-seeking behaviours related to diabetes, weight loss, physical activity, smoking cessation, and medication adherence [21]. Chatbots are a type of mHealth strategy that builds interactivity into digital interventions. By enabling two-way communications, behaviourally-informed chatbots offer a dynamic format to deliver strategies known to encourage behavioural change more effectively. However, few digital interventions for violence prevention have been tested to date in African settings [16].

Edutainment is another promising approach to reduce young women's violence exposure, though this has been understudied to date. Through relatable characters or influential messengers, edutainment can convey educational messages, relevant information [such as where to seek help], or promote healthful social norms [22].

The evidence for whether edutainment can reduce IPV exposure or improve gender attitudes is promising, but inconclusive. *Soul City*, a large-scale edutainment campaign in South Africa, led to knowledge about IPV, improved self-efficacy, and increased helpline calls [23]. The *MyPlan* app in Kenya improved women's safety strategies but had null effects on violence exposure [24]. In Uganda, a media campaign in 112 villages featuring video vignettes during community film festivals significantly reduced IPV but had null effects on attitudes towards IPV [25]. In Egypt, short videos delivered via *Whatsapp* or *Facebook* detected no improvements in violence exposure or gender beliefs but did increase women's knowledge and intention to access resources in future cases of IPV [26].

We conducted a trial testing whether two versions of a behaviourally-informed chatbot could improve gender attitudes in order to reduce IPV exposure among young women in South Africa.

## Materials & methods

We conducted a digital, individually randomised controlled trial with 4 arms, consisting of:

- Pure Control: quizzes measuring outcomes only

- Attention Treatment: interactive chatbot featuring information about sexual health

- T1: gamified interactive chatbot delivering behaviourally-informed content

- T2: narrative-based interactive chatbot delivering behaviourally-informed content

The target audience for *ChattyCuz* was young women aged 18 to 24-years-old. Our primary hypotheses are as follows: If chatbots can be effective at improving attitudes and relationship safety among young South African women, compared to pure control T1 and T2 will demonstrate:

1. higher mean score on gender attitudes scale, and;

2. lower women's past-month self-reported IPV exposure

We secondarily hypothesised that identification of unhealthy relationship behaviours would be higher and depressive symptoms would be lower for T1 and T2 compared to control.

## Inclusion criteria

We targeted a sample of South African young women who responded to targeted advertising on Facebook, inviting them to engage with a *WhatsApp* chatbot. Advertising prompts were sent only to individuals who self-identify on their *Facebook* profiles as female between 18 and 24 years; and were living in South Africa (though they may be national or non-national residents).

## Treatment arms

*Pure control arm* exposed eligible participants only to specific evaluation quizzes. The quiz pertaining to gender attitudes was delivered directly after eligibility screening. Follow-up quizzes three months later evaluated IPV exposure and the brief depressive symptoms screener.

*Attention treatment arm (T0)* was modelled on an existing chatbot, *Big Sis*. It featured the *ChattyCuz* brand and persona, but did not include any of the behaviourally-informed content modules. Instead, the Attention Treatment chatbot addressed the basics of sex, reproduction and puberty, and allowed users to browse topics by thematic areas, such as "love and relationships" (e.g. "Do relationships = sex?") "sex basics" (e.g. "What is virginity?", "Why might sex hurt?") and "protecting yourself" (e.g. "How to use a condom"). The content comprised overlapping concepts and thematic areas, which enabled the delivery of the same evaluation quizzes as the treatment arms.

*ChattyCuz-Gamified (T1)* is a WhatsApp chatbot that aims to support young women in navigating intimate relationships. *ChattyCuz* is designed as a safe, private space. It offers simple, symbolic rewards to guide and motivate users throughout their interactions. T1 follows an interactive script that adapts depending on responses provided by the user. Regardless of whether the user answers a question correctly, *ChattyCuz* gives feedback on the response. T1 was designed to appear as a unique personality based on the characteristics of an older cousin (approachable, relatable, and supportive of the user). She takes users through a series of different topics, offering a critical reflection on power in relationships; skills practice to identify unhealthy relationship behaviours; learn healthy communication and coping mechanisms; affirming values to build self-efficacy; and planning for safety. Users are encouraged to collect six Squad Members to become a Relationship Superstar (Table 1).

*ChattyCuz-Narrative (T2)* uses storytelling to connect with audiences and delivers the themes through *WhatsApp* voice notes (short digital audio clips) and short sections of text. The narrative follows the life of Sandiswa, a young woman engaged in an embattled relationship with a man named Jabu. Each voice clip opens with a short, ambiance-rich scene (two characters arguing; a character at a family party with her sister-in-law; a character playing a song). The scene then fades and Sandiswa speaks directly to the user to summarise what happened, how she feels and the dilemma she faces. By 'helping' Sandiswa to make life choices (who she should trust, how to communicate with Jabu, and how to plan for safety) ChattyCuz-Narrative's design aims to improve users' knowledge and relationship skills in line with module aims. The aim of this format is to make the user feel as if she is a trusted person that the narrator wants to speak to during her daily life. To maximise accessibility of the narrative, the user is presented with the choice of hearing the story through voice clips, or reading it as text.

Additional information about each chatbot can be found in the Online Supplemental Appendix.

**Table 1. Module content for gamified chatbot and narrative chatbot.**

| Module | Challenge | Module aim |
|---|---|---|
| | *I have...* | *I can...* |
| **Module 1** | Limited awareness of unhealthy relationship behaviors | Identify unhealthy relationship behaviors |
| **Module 2** | Limited social support | Identify a trusted person to speak to about an unhealthy relationship |
| **Module 3** | Low autonomy and self-efficacy | Improve self-efficacy by engaging in self-affirmation |
| **Module 4** | Limited understanding of healthy communication skills in my intimate relationships | Practise healthy communication skills for relationships |
| **Module 5** | An action-intention gap preventing safety seeking | Make a safety plan |
| **Module 6** | Limited psychological support during difficult experiences | Identify practical coping mechanisms for difficult experiences |
| **All Modules** | Limited awareness of power imbalance and control in relationships | Reflect critically on power imbalance and control in relationships |
| **Gamified Chatbot rationale** | Gamification–enhancing a service or activity with a motivational reward to provoke a specific knowledge-based, behavioral, or attitudinal outcome [39]—has promise for participant engagement and propensity to learn from their experiences. There is some evidence to suggest that timely, structured feedback can improve one's ability to successfully complete logical tasks, such as mathematics tests [40]. Game designs have been successfully applied to digital sexual and reproductive health interventions, such as interactive computer programs aimed to enhance adolescents' sense of self-efficacy, and digital games designed to improve adolescents' knowledge [41]. | |
| **Narrative Chatbot rationale** | Narrative-based edutainment can also have positive effects on sex and relationship behaviors. A study of the phased rollout of a soap opera promoting family planning in Brazil points to the show's effect on people's choices: areas where the show was broadcast recorded lower fertility rates, and changes in naming conventions.[42] In Nigeria, exposure to *MTV Shuga* –a drama featuring educational storylines about HIV/AIDS–has been shown to improve viewers' knowledge and attitudes towards HIV and reduce women's prevalence of sexually transmitted infections [43]. In a quasi-experimental study in Brazil, the multimedia narrative *Program H* have been shown to improve knowledge, attitudes, practices related to sexual health, and reduce gender-based violence [44]. | |

## Measures

The outcome variables for this trial are adapted versions of existing validated scales translated into quizzes (mimicking those in teen magazines) through an iterative, user-centred process. We conducted cognitive interviews to test quiz questions with end users, gathering their feedback on question phrasing and level of comprehension. In October 2020, we validated multiple, short-form adapted surveys ($n$ = 1,088 young women), and invited a sub-sample to retest in December 2020 ($n$ = 231).

*Gender attitudes* were measured 4 days after allocation using an adapted *Gender Relations Scale* (aGRS) [27]. The aGRS is comprised of 6 items scored on Likert-type responses and had an acceptable internal consistency in the pre-trial validation with $n$=1,088 participants (Cronbach's α = 0.75). An example item is "It's natural for a guy to have girls on the side while he's dating you." We analysed aGRS as a continuous outcome with higher scores denoting more gender equitable beliefs.

*IPV exposure* was assessed 3 months after allocation using an adapted *WHO Multi-country Study Instrument* (aWHO) [28]. comprising two physical and one sexual violence measure shown to be sensitive and specific in the South African setting. An example item is "In the past month, did your bae ever slap, hit you or throw something at you?". In addition, we added two technological violence items. An example is "If you sent your bae a sexy photo of yourself, are you afraid they might share it?" In the validation assessment (n = 67), internal consistency of the aWHO was acceptable (Cronbach's α = 0.75).

*Identification of unhealthy relationship behaviors* was measured 3 days after allocation using an adapted *Intimate Partner Violence Attitudes Scale* (aVAS) [29]. This scale comprises abuse and control subscales to measure psychological violence identification. An example item is "My bae never hurts me but I understand why sometimes he threatens to do it so that I get in line." Validation with 74 participants found that internal consistency was high (Cronbach's α = 0.82).

*Depressive symptoms* were assessed using a brief self-reported symptom screener: the two-item *Patient Health Questionnaire* (PHQ-2). A score of 3 or greater is suggestive of probable depression.

*Socio-demographic* variables were ascertained in single item form at baseline. Age was self-completed. Gender was self-reported as either female, male, or non-binary. Sexual orientation was ascertained through a single item asking users whether they were attracted to men or women, "it is complicated" or "prefer not to say". Relationship status was assessed through a single item asking whether users were seeing "someone special". The state of participants' mental distress was assessed through one item asking if, in the past month, they had been mostly "energised and hopeful", "just getting along", or "down and hopeless". Food security level was assessed by an item asking if users had been hungry in the last two months because "there was not enough food in their house". Household composition was measured using a single item asking how many people lived in the household with the user. Participants were also asked whether they had any children. Province was self-reported.

## Procedures

Facebook advertisements targeted mobile devices (to be viewed on mobile news feeds only) and seamlessly directed interested participants to *WhatsApp*. We conducted pipeline randomisation using a custom-built randomisation algorithm implemented as an R application and run as a separate web service. After consenting to participate in the study, users were randomised into either Pure Control (20%), Attention Control (20%), ChattyCuz-Gamified (T1), or ChattyCuz-Narrative (T2).

The trial was set up on RapidPro (open source jointly owned by UNICEF and Nyaruka), an open-sourced user management system that records user states and manages inbound and outbound messages between users and the chatbot [30]. The trial was set up with content stored in the form of RapidPro "flows", a series of steps with built-in logic to direct users through the experience based on their interactions with the chatbot.

When users were allocated into the different trial arms using the randomisation algorithm, users' records were stored in a PostgreSQL database, along with user data and message interactions exchanged between users and *ChattyCuz*. To support chatbot users, careful monitoring of the database was required as it grew throughout the trial (possibly due to the design of the program, which involved a high ratio of outbound to inbound messages).

Users participating in the trial were remunerated with ZAR 30 (approximately $2.00) on completion of all components of the respective trial arms, and ZAR 10 (approximately $0.70) on completion of a follow-up quiz three months later. Remuneration was carried out by connecting RapidPro to a DT One account with pre-loaded airtime. This integration with DT One is built into RapidPro and requires only the DT One account and API key to be specified to connect the account. The airtime remunerations were defined based on local research standards and triggered automatically when users reached the end of specified RapidPro flows within the different trial arms.

## Data privacy and protection

Personal data collected in the trial was covered by Praekelt.org's data protection policy. Data handling, storage and transfers comply with national and international guidelines, standards and legal frameworks, notably the General Data Protection Regulation of the European Union. *ChattyCuz* complies with the record management in the Protection of Personal Information Act 4 of 2013.

User data recorded during the trial included age, gender and telephone number, and responses to quizzes, polls or surveys, or questions asked by the user. This data was stored in a

secure PostgreSQL database. Users were exposed to Terms and Conditions as part of the consent procedure (S1 Table). The Terms and Conditions explained, in simple terms, that should they choose to participate, their data would be collected and used for statistical or research purposes. This data was visible to both Praekelt.org and BIT throughout the trial. All personally identifiable information was removed from the data (i.e. the data was anonymised) before we transferred it from Praekelt.org to BIT for analysis.

## Consent procedures

During the onboarding process, users received information about the intervention and the research study. After users clicked on a Facebook advertisement on their device, they were directed to WhatsApp. On WhatsApp, users were exposed to a brief description of the trial with all the key information required for consent. The user then received a PDF explaining the nature of the trial in plain terms. The user was asked if they wanted to participate in the trial; if they agreed, they were allocated to a trial arm at random using a built-in algorithm. The user was automatically remunerated with preloaded airtime after the completion of the primary outcome quiz (S1 Fig). Users who could not receive automated airtime remunerations were sent money vouchers, redeemable as cash or airtime at any bank ATM, manually via SMS.

## Analysis

For primary analysis 1, we estimated the following model:

$$\gamma_i = \alpha + \beta_1 T1_i + \beta_2 T2_i + \beta_3 BAU_i + X_i\beta_4 + \mu_i$$

Where:

- $\gamma_i$ is Primary Outcome 1; a score of attitudes and beliefs on power equity in relationships;

- $T1$ is a dummy variable = 1 if individual $i$ was allocated to T1 (Gamified bot);

- $T2$ is a dummy variable = 1 if individual $i$ was allocated to T2 (Narrative bot);

- $BAU$ is a dummy variable = 1 if individual $i$ was allocated to Attention control;

- $X_i$ is vector of control variables, which includes:

  ○ A measure of Baseline Attitudes and Beliefs on Power equity in the relationships (taken at the beginning of the chatbot experience).

  ○ Region;

  ○ Gender;

  ○ Age;

  ○ Mental health score;

  ○ Socioeconomic status—food security;

  ○ Household composition;

  ○ Children;

  ○ Partnership status;

  ○ Province.

- $u_i$ is the error term.

The coefficients of interest are $\beta_1$, which measures the impact of being allocated to T1 compared to Control; $\beta_2$, which measures the impact of being allocated to T2 compared to Control; and $\beta_3$, which measures the impact of being allocated to Attention Control compared to Control, on the outcome measure of interest.

We estimated the model using ordinary least squares and used heteroskedasticity robust standard errors. We modelled the primary outcome with a series of three regressions: (1) an unadjusted treatment-only model, (2) a model adjusted for covariates gender and region of recruitment, (3) a model adjusted for a full set of collected covariates. The models will be compared using a goodness-of-fit statistic, AIC. If there are qualitative differences between the results of these models, we will focus our reporting on the results of the model with the lowest AIC.

We adopted a strategy of using multiple comparisons adjustments for the primary analysis of primary and secondary outcomes, employing the Benjamini-Hochberg step-up procedure. In brief, this uses the traditional $p < 0.05$ threshold as a marker of statistical significance, but reduces it each time an additional comparison is made. The primary analysis adjusts for 2 comparisons: T1 vs. Pure Control; T2 vs. Pure Control. The primary analysis for secondary outcomes adjusted for 2 comparisons: T1 vs Attention, and T2 vs Attention.

### Ethical considerations

Our trial received ethical approval by Pharma-Ethics (Protocol number 2019063; 21 September 2020). We pre-registered our trial on the Open Science Framework (16 March 2021).

Given the sensitive nature of this study, the safety for the study population informed the design of the trial and the platforms through which it was delivered. Participants in the trial were 18 or older, the age when young persons are able to consent on their own to research in South Africa. For participants requiring additional support, access to text-based individual counselling services was provided by an external counselling service provider, *MobieG*. This non-governmental organisation already offered text-based counselling, and our trial provided additional training specific to responding to IPV.

We established a platform for the trained *MobieG* counsellors to use *WhatsApp* for individual counselling. Reply messages from counsellors were channelled through *RapidPro* without interrupting the active conversations between users and counsellors. A benefit of this approach is that it allowed us to keep the interactions with the counsellors within WhatsApp, rather than directing users to an external platform that would have disrupted the user experience. A total of $n = 631$ users took up the counselling (4·5% of the sample who were followed to midline). The quality of the counselling experience and moderation of trial outcomes will be the subject of a separate paper.

### Results

We recruited 19,643 women in 31 days. As expected, 20.0% were allocated to Pure Control, 20.0% to Attention Control, 30.0% to *ChattyCuz-Gamified* and 30.0% *ChattyCuz-Narrative* (Fig 1).

As Table 2 indicates, baseline characteristics are very similar for all these conditions. Across all arms, the average age was 21.2 and 70.7% were in the relationship. Overall, 31.7% of participants reported to have symptoms consistent with probable depression, with similar rates across arms.

Of the 19,643 participants we recruited, 11,630 (59.2%) completed the Adapted GRS quiz, our first primary outcome. We refer to these participants as midline participants. As expected, almost all (98%) of participants in the Pure Control completed the midline quiz as this was

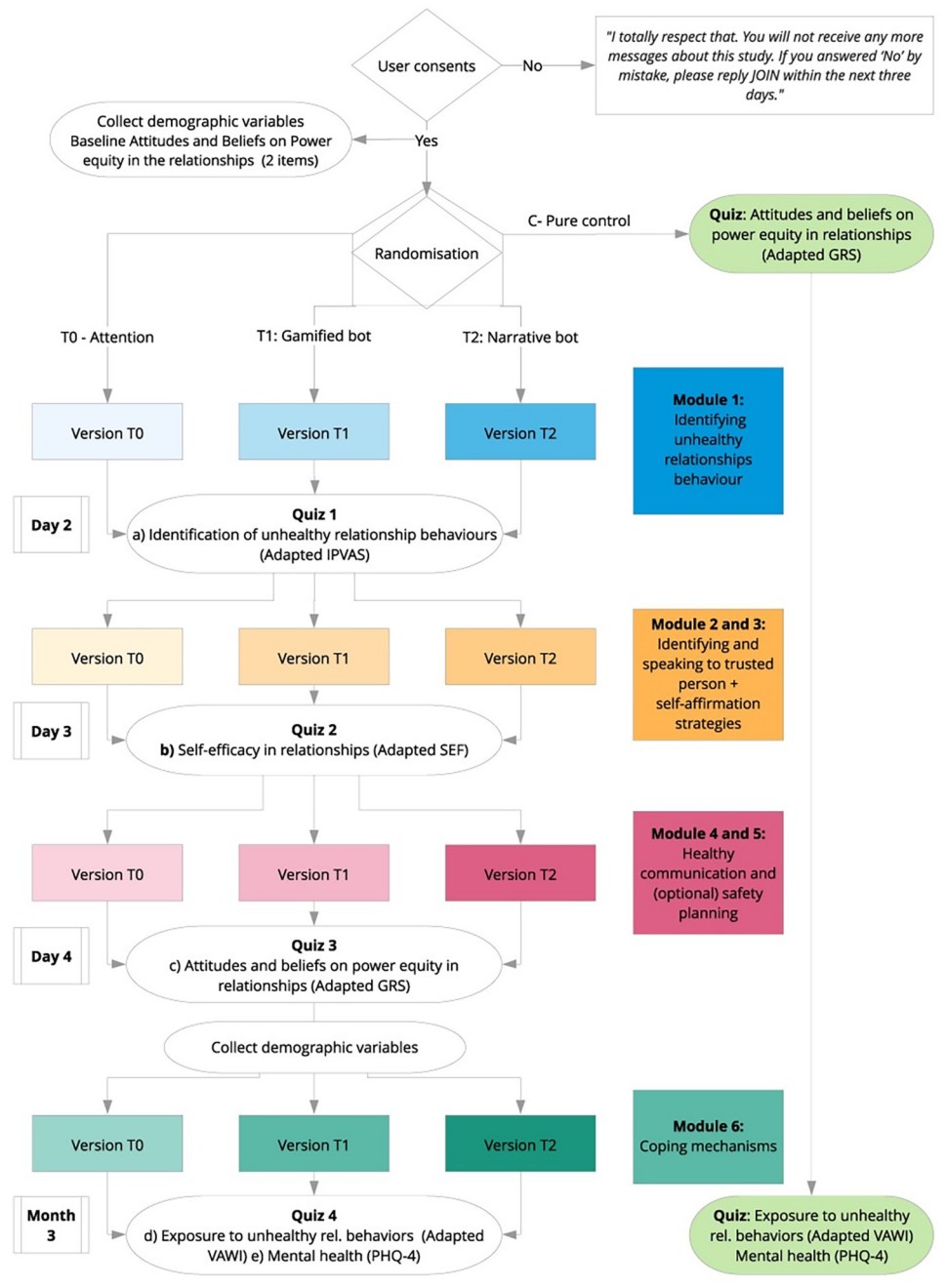

**Fig 1. Trial flow diagram.**

asked on the same day as recruitment in this trial arm. There were differences between trial arms regarding retention of participants, with T0 having notably high attrition. The baseline covariates are similar across all trial arms. However, in terms of statistical significance, the treatment groups are slightly different. Descriptive statistics for all groups and highlights statistically significant imbalances, where groups differ on attributes at the 10% level confidence level, in red (Table 2). These differences are due to different rates of attrition and perhaps different types of attrition across the treatment groups.

**Table 2. Participant socio-demographics by trial arm.**

| | Baseline | | | | Midline (4 days after allocation) | | | | Endline (3 months after allocation) | | | |
|---|---|---|---|---|---|---|---|---|---|---|---|---|
| | Pure Control | Attention T0 | Gamified T1 | Narrative T2 | Control | T0 | T1 | T2 | Control | T0 | T1 | T2 |
| Proportion followed | - | - | - | - | 98.2% | **24.0%** | 60.6% | 56.5% | 38.1% | 39.4% | 32.0% | 31.4% |
| Participants | 3,930 | 3,929 | 5,891 | 5,893 | 3,867 | 928 | 3,517 | 3,318 | 1,474 | 366 | 1127 | 1043 |
| Average age | 21.2 | 21.2 | 21.2 | 21.2 | 21.2 | 21.2 | **21.1** | **21.1** | 21.4 | 21.3 | **21.2** | 21.3 |
| In current relationship | 70.7% | 70.7% | 70.7% | 70.7% | 70.9% | 73.2% | 70.0% | 70.5% | 75.1% | **80.1%** | 75.9% | 77.1% |
| Mental distress at baseline | 31.7% | 31.8% | 31.7% | 31.7% | 31.7% | **28.8%** | 30.5% | 30.8% | 30.9% | 29.0% | 29.3% | 30.5% |
| Gender attitudes at baseline 1* | - | - | - | - | 0.00 | 0.00 | **0.04** | **0.07** | -0.20 | -0.10 | -0.10 | -0.10 |
| Gender attitudes at baseline 2 * | - | - | - | - | 0.00 | **0.08** | **0.05** | **0.08** | 0.00 | -0.10 | 0.00 | 0.00 |
| Food insecure | - | - | - | - | - | - | - | - | 54.2% | 55.2% | 51.2% | 52.3% |
| Members in household | - | - | - | - | - | - | - | - | 3.9 | 4.0 | 3.9 | 4.0 |
| Has 1+ child | - | - | - | - | - | - | - | - | 41.8% | 38.0% | 43.7% | 39.4% |

\* Gender attitudes at baseline normalized by subtracting the Pure Control mean from the mean, and dividing by the control group standard deviation; higher scores denote stronger beliefs in power equity

Bold text denotes balance checks indicate greater than 10% variation in socio-demographic

## Primary outcome: Equitable attitudes

We found that all three chatbots improved beliefs in power equity in relationships compared to receiving no intervention ("pure control"). The effect sizes for the Attention Control, Narrative and Gamified bot were 0.10, 0.20 and 0.29 (Cohen's *D*, 2.s.f.) respectively. While these effect sizes are modest, they are substantial in terms of real-world significance given the potential scale at which chatbots can be delivered.

Effect sizes for the experimental chatbots (Narrative and Gamified) were found to be relatively stable after adding important controls to the models (gender attitudes, age, mental health, and partnership status at baseline). Effect sizes for the Attention Control, however, decreased as more controls were included, confirming the suspicion that attrition and differences in observed variables could account for the treatment effects observed in this trial arm.

**Table 3. Primary trial outcomes, by randomization arm: Gender beliefs and partner violence.**

| | Gender attitudes | | | Past-month IPV | |
|---|---|---|---|---|---|
| | Unadjusted | Adjusted^ | Adjusted^ | Unadjusted | Adjusted^ |
| | OLS | OLS | CEM | OLS | OLS |
| **Attention Control (T0)** | 0.70** | 0.50** | 0.43* | - | - |
| | -0.2 | -0.2 | -0.2 | | |
| **ChattCuz Gamified (T1)** | 1.66** | 1.49** | 1.50** | -0.06** | -0.07** |
| | -0.1 | -0.1 | -0.1 | -0.02 | -0.02 |
| **ChattCuz Narrative (T2)** | 1.08** | 0.99** | 0.96** | -0.02 | -0.02 |
| | -0.1 | -0.1 | -0.1 | -0.02 | -0.02 |
| **Observations** | N = 11,630 | N = 11,519 | N = 11,447 | N = 4139 | N = 3955 |

+ p < 0.10

\* p < 0.05

\*\* p < 0.01

^ additional controls are: baseline gender attitudes, age, mental health at baseline, partnership status, food security, household composition, number of children

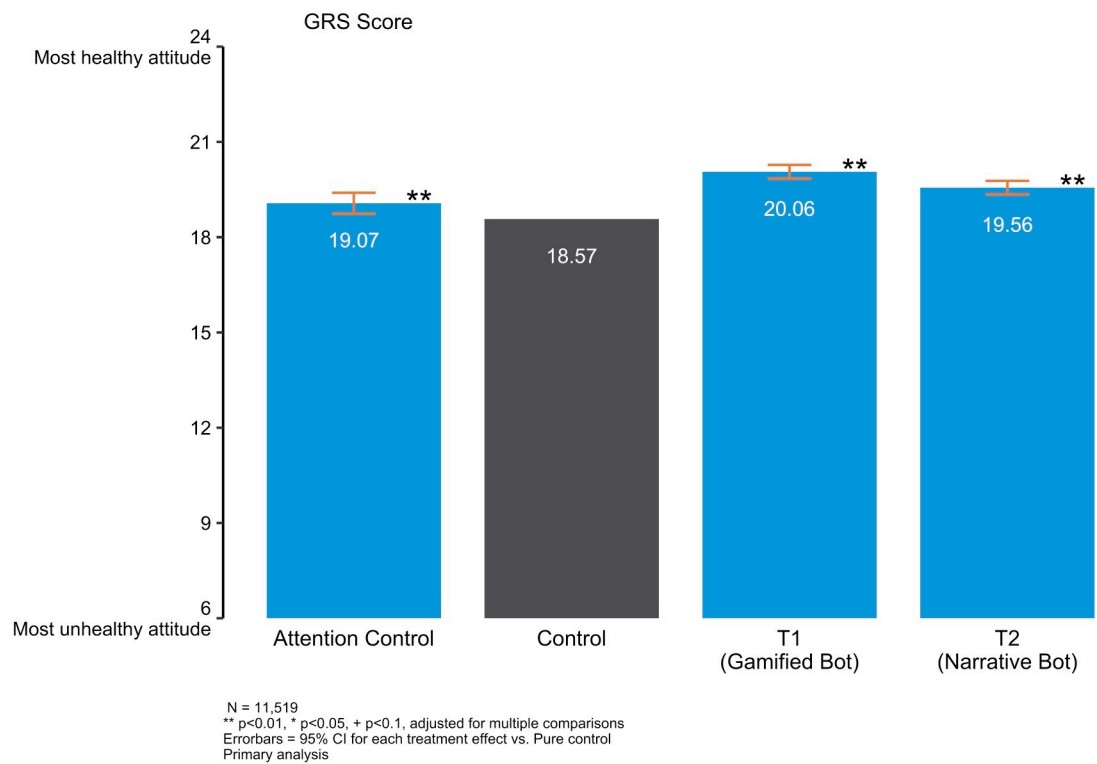

**Fig 2. Gender relations scale (T1, T2, and Attention Control compared to Control).**

As indicated in Table 3, we find the effect of all chatbots are significant at 5% level across specifications, after correcting for multiple comparisons.

Fig 2 presents the outcome for regression 3, the prespecified model.

## Primary outcome: IPV Exposure

We found small but significant effects for T1 on overall past-month IPV exposure at three-month follow-up (Table 3). The 1043 women who completed the follow-up quiz from the *ChattyCuz-Gamified* arm reported lower rates of IPV exposure (55%) than young women in the pure control group (62%, Fig 3).

T2 demonstrated no effects on IPV exposure. Estimates for Attention Control were not robust to additional controls, suggesting differential attrition based on observed variables account fully for treatment effects observed in this arm. We therefore omit T0 in the presentation of data.

In supplemental analysis we examine IPV by type and see that all types of violence are reduced, with psychological and technological forms of IPV significantly reducing this exposure by 8% (S2 Table). Physical and/or sexual IPV was reduced by 2% and this point estimate did not reach statistical significance (S3 Table).

## Secondary outcomes

**Identification of unhealthy relationship behaviours.** We found that both T1 and T2 increased users' ability to correctly identify unhealthy relationship behaviours compared to the Attention Control. On a scale of 6–18 the score of the Attention Control group was 14.80 while the score of the T1 and T2 chatbots were 15.51 and 15.22 respectively (Fig 4).

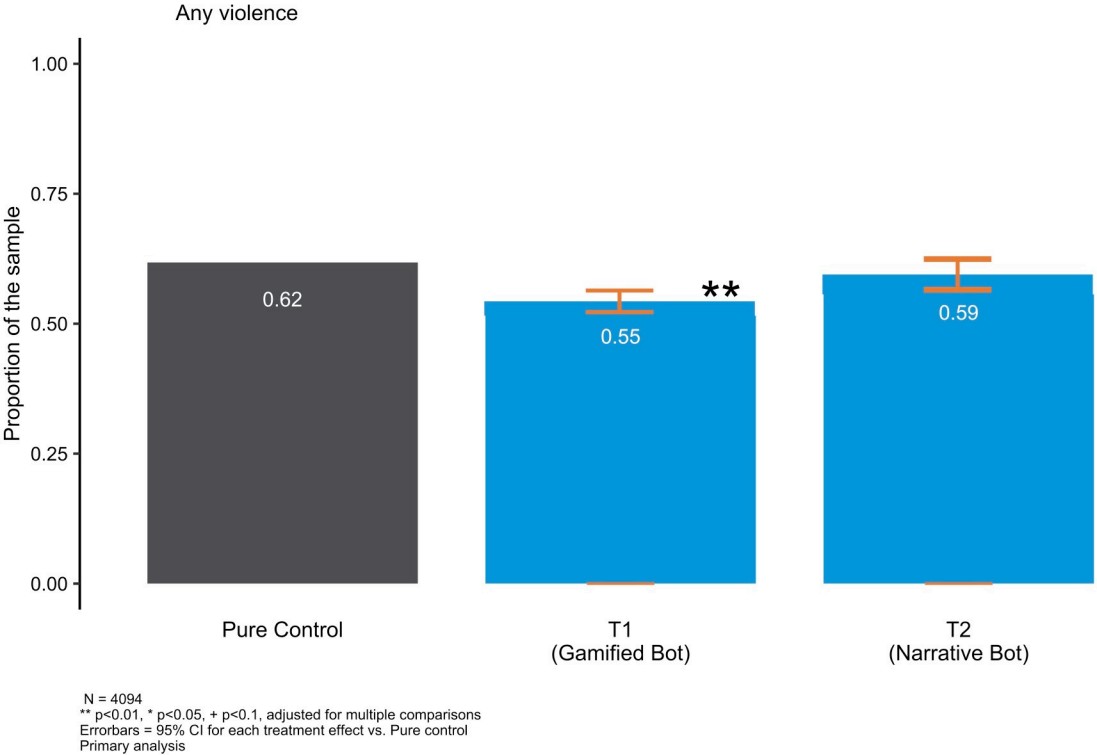

**Fig 3. Past-month IPV exposure (T1 and T2 compared to Control).**

### Mental health

T1 was slightly protective for depressive symptoms, but this was not statistically significant (S2 Fig).

The proportion of users who expressed interest in receiving counselling was much higher in the control conditions (above 17%), compared to 8.4% of users in the *ChattyCuz-Narrative* group and 6.9% in the *ChattyCuz-Gamified* group. Similarly, users in the control conditions accessed counselling at more than twice the rate of the treatment arms.

## Discussion

In an experimental trial among young women in South Africa, a gamified digital chatbot improved equitable gender beliefs and made a small, but significant, reduction in past-month IPV exposure. This gamified chatbot is among the first digital interventions to effectively reduce IPV exposure among young women. The chatbot significantly improved skills around identifying unhealthy relationships, but did not alter depressive symptoms compared to the control. These primary results are robust across specifications and robustness checks, including comparison to an attention control condition.

Ours is the first study, to our knowledge, to compare two digital communication approaches in relation to actual IPV exposure that offers depth to the growing edutainment field. We hypothesise that the observed reduction in IPV is due to the gamified chatbot offering tailored, immediate feedback on gender beliefs, identifying unhealthy relationship behaviours, and implementing skills to stay safe–all shown to be precursors for staying safe in relationships [13,31,32]. The *MyPlan* app in Kenya had promising results around safety skills and intentions, but no effect on violence exposure [24].

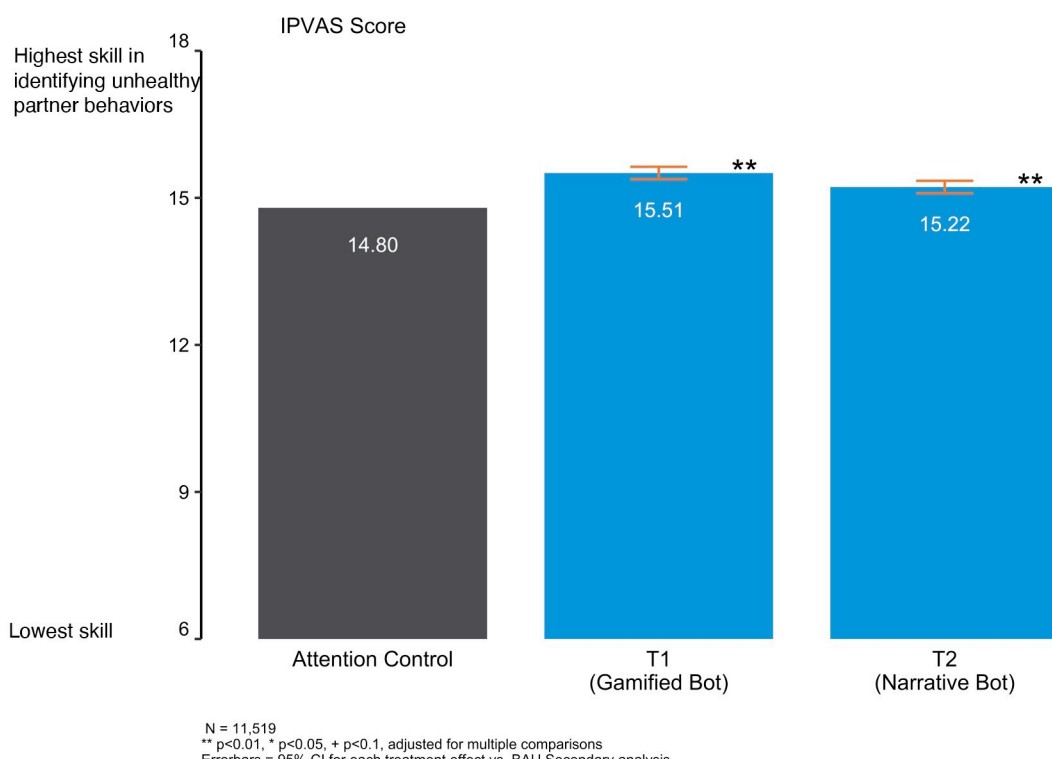

Fig 4. **Identification of unhealthy relationship behaviors (T1 and T2 compared to Attention Control).**

While the narrative chatbot did improve gender equitable attitudes, it had no measurable effect on IPV exposure. Two hypotheses may frame these null findings. Firstly, the narrative chatbot was less directive, allowing users to reach their own conclusions from the experience of a fictional character. In India, "explicit" calls to action were more impactful than "implicit" narratives in terms of changing beliefs about IPV [33]. Similarly, the effective intervention using mass media in Uganda reduced IPV not by shifting attitudes [as these remained largely unchanged] but by increasing women's willingness to report violence [25]. Secondly, since many young South Africans already have a high saturation of narrative programming through *Soul City* [23], which may offer a ceiling effect in this particular setting.

Overall, both behaviourally-informed chatbots were significantly better at encouraging user retention and engagement, compared to information-focused chatbots. Technological solutions may improve youth's access to care and could complement existing in-person services if young people feel more engaged or less threatened with this medium of communication [34]. Only 24% of Attention Treatment participants were retained (compared to 56% and 60% for T1 and T0, respectively). This could indicate that the behaviourally-informed chatbots are more engaging than typical information-only digital approaches. On the other hand, the high attrition limits our ability to assess impact on IPV exposure for the Attention Treatment arm.

The study was not powered to detect distinctions by IPV type, but this sub-analysis is important for the field. The gamified chatbot most effectively decreased reported exposure to technological and psychological safety. This has relevance for young women's wellbeing, since technological and psychological forms of IPV may be more common, and have demonstrated preliminary causal associations with mental and physical health [35,36]. Physical and/or sexual IPV trended towards decreased exposure (*p*=0.02*)*, suggesting this may require more targeted or in-person approaches.

Prevalence of depressive symptoms among this relatively large sample of young women was high, with 32% reporting symptoms consistent with probable depression on a brief screener. This rate is similar to studies nationally, which find up to 41% of young women exhibit elevated depressive symptoms [37]. Neither experimental chatbot was effective at reducing depressive symptoms, nor were they associated with increased distress. This offers an important confirmatory finding for the IPV field, since one ethical consideration for working with younger populations or intervening through low-cost digital interventions is the risk that disclosure of IPV could cause additional distress or even psychological harm. Our findings refute this claim and suggest that—with appropriate links to online counselling services—technologies like chatbots may neither increase depressive symptoms, nor be protective.

## Implications for research and practice

The *WhatsApp* application is a popular internet protocol messaging platform, providing an opportunity to deliver chatbots at greater scale as the market expands. There are, however, inherent risks of digital IPV interventions, particularly if phones are not private or if male partners use technological violence or other controlling behaviours. Our opt-in, live individual counselling delivered via *WhatsApp* was an important safety consideration of the trial. This safety enhancement was able to draw upon a South African organisation with expertise in chat-based counselling for young people–a partnership that may be harder to identify in other LMIC settings. Providing this counselling service at-scale could be cost-effective (if it serves to identify and refer roughly 4.5% of those at highest need for individualised care), but requires a more integrated approach to de-identifying user data.

Several potential avenues for future research exist. Primary prevention with younger adolescent women, male peers, and adolescent pregnant women seems most promising given the high exposure to violence victimisation and perpetration among these groups. *ChattyCuz* was designed to reach girls at a young age, though South African ethical constraints mandate that research working directly with participants occur over the age of 18. Future studies should adapt and pilot the intervention for adolescents aged 17 or younger, finding creative methods of working with ethics boards to waive the need for parental informed consent [which negate the chatbot potential to reach young persons directly and may itself lead to safety concerns]. It would be important to evaluate whether this intervention could have a similar, or even more significant, effect among younger girls if we reach them prior to sexual debut or while forming initial intimate relationships.

## Limitations

Findings should be viewed in light of study limitations. We recruited young women who were already on *Facebook* with access to a mobile phone with internet connection. While *Facebook* is one of the most widely used platforms, we are unable to generalise findings to young women with no private phone access or who do not use *WhatsApp* or *Facebook*. This is an important sampling limitation, since those young people who are highly vulnerable to violence exposure may have limited access to digital technologies. The heteronormative framing of the intervention is imprecise and deserves increased attention in South Africa, where sexual and gender minorities are at higher risk of IPV exposure than heterosexual counterparts [38].

Several design characteristics of digital intervention research are worth noting as potential limitations. Low literacy may have inhibited the ability of some participants to take part in the intervention or quizzes. We chose not to measure education as a socio-demographic covariate, but future studies should include this. It is possible that participants completing the quizzes were different from those who filled out baseline surveys, but we chose not to ask for South

African identity numbers or private details because of confidentiality risks. Attrition can only be partially accounted for in our analytical approach. While retention was better than for most digital interventions, it is plausible that those in the endline sample are systematically different across arms. Indeed, we find that in the endline sample, there are statistically significant differences in observable characteristics between subgroups. However, because our models are able to control for these measurable differences, differential attrition due to observed characteristics should not affect the overall results. However, we are unable to control for unobserved characteristics and our main analysis requires a missing at random assumption. We conduct additional "robustness checks" in response to concerns that data is missing not at random, including a sensitivity analysis.

## Conclusion

Digital solutions represent an important approach to scale efforts to reduce IPV among young women. A behaviourally-informed chatbot improved gender equitable attitudes and reduce exposure to IPV in the short-term. When coupled with tailored support for a small number who would benefit from live online counsellors, behaviourally-informed chatbots have potential to reach a large subset of population at low-cost. Chatbots may offer a promising strategy to support healthy norms and safe relationships.

## Supporting information

**S1 Fig. Consent flow.**
(DOCX)

**S2 Fig. Mental health by trial arm, n = 5,022.**
(DOCX)

**S1 Table. Intervention modules.**
(DOCX)

**S2 Table. Gender equitable beliefs regression.**
(DOCX)

**S3 Table. Any IPV exposure regression.**
(DOCX)

**S4 Table. Technological and/or psychological IPV.**
(DOCX)

**S5 Table. Physical and/or sexual IPV.**
(DOCX)

## Acknowledgments

We are appreciative in the many young women who took part in user testing, cognitive interviews, the pilot, formative co-creation workshops, and the Community Advisory Board. We thank Nkeletseng Tsesane for supporting the Community Advisory Board recruitment and Lele van Eck for facilitating their group discussions. We thank the MobieG team for direct counselling support for participants who required additional help. Girl Effect was monumental to the design of the gamified chatbot and the attention condition was composed of their Big Sis chatbot. The research team at BIT supported design and analysis.

## Author Contributions

**Conceptualization:** Alexandra De Filippo, Paloma Bellatin, Eli Grant, Puseletso Nkopane, Camilla Devereux, Benjamin Vermeulen.

**Data curation:** Paloma Bellatin, Neville Tietz, Alexander Whitefield, Kaitlyn Crawford.

**Formal analysis:** Paloma Bellatin, Alexander Whitefield.

**Funding acquisition:** Alexandra De Filippo, Paloma Bellatin, Eli Grant, Benjamin Vermeulen.

**Investigation:** Alexandra De Filippo, Paloma Bellatin, Neville Tietz, Eli Grant, Alexander Whitefield, Puseletso Nkopane, Camilla Devereux, Kaitlyn Crawford, Abigail M. Hatcher.

**Methodology:** Alexandra De Filippo, Paloma Bellatin, Neville Tietz, Eli Grant, Alexander Whitefield, Camilla Devereux, Benjamin Vermeulen, Abigail M. Hatcher.

**Project administration:** Paloma Bellatin, Neville Tietz, Puseletso Nkopane, Kaitlyn Crawford, Benjamin Vermeulen, Abigail M. Hatcher.

**Resources:** Neville Tietz, Benjamin Vermeulen.

**Software:** Neville Tietz, Benjamin Vermeulen.

**Supervision:** Abigail M. Hatcher.

**Validation:** Alexandra De Filippo, Paloma Bellatin, Alexander Whitefield, Puseletso Nkopane, Camilla Devereux, Benjamin Vermeulen.

**Visualization:** Alexander Whitefield, Abigail M. Hatcher.

**Writing – original draft:** Alexandra De Filippo, Paloma Bellatin, Abigail M. Hatcher.

**Writing – review & editing:** Alexandra De Filippo, Paloma Bellatin, Neville Tietz, Eli Grant, Alexander Whitefield, Puseletso Nkopane, Camilla Devereux, Kaitlyn Crawford, Abigail M. Hatcher.

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
