## [Decision Letter · Decision Letter 0]

9 May 2023

PDIG-D-22-00354

Effects of digital chatbot on gender beliefs and exposure to intimate partner violence among young women in South Africa

PLOS Digital Health

Dear Dr. Hatcher,

Thank you for submitting your manuscript to PLOS Digital Health. After careful consideration, we feel that it has merit but does not fully meet PLOS Digital Health's publication criteria as it currently stands. Therefore, we invite you to submit a revised version of the manuscript that addresses the points raised during the review process.

Please submit your revised manuscript within 30 days Jun 08 2023 11:59PM. If you will need more time than this to complete your revisions, please reply to this message or contact the journal office at digitalhealth@plos.org. We look forward to receiving your revised manuscript.

Kind regards,

Laura M. König

Academic Editor

PLOS Digital Health

Journal Requirements:

2. We ask that a manuscript source file is provided at Revision. Please upload your manuscript file as a .doc, .docx, .rtf or .tex.

3. Please provide separate figure files in .tif or .eps format.

Additional Editor Comments (if provided):

Reviewers' comments:

Reviewer's Responses to Questions

**Comments to the Author**

1. Does this manuscript meet PLOS Digital Health’s publication criteria? Is the manuscript technically sound, and do the data support the conclusions? The manuscript must describe methodologically and ethically rigorous research with conclusions that are appropriately drawn based on the data presented.

Reviewer #1: Yes

Reviewer #2: Yes

2. Has the statistical analysis been performed appropriately and rigorously?

Reviewer #1: Yes

Reviewer #2: Yes

3. Have the authors made all data underlying the findings in their manuscript fully available (please refer to the Data Availability Statement at the start of the manuscript PDF file)?

Reviewer #1: Yes

Reviewer #2: Yes

4. Is the manuscript presented in an intelligible fashion and written in standard English?

Reviewer #1: Yes

Reviewer #2: Yes

5. Review Comments to the Author

Reviewer #1: I would like to thank the publisher and the journal for involving me in the peer review of this very interesting study on the “Effects of digital chatbot on gender beliefs and exposure to intimate partner violence among young women in South Africa”.

The study is conducted with great scientific attention and deals with a level of prevention and promotion of crucial importance for people's health and particularly for young women who may be victims of IPV. However, in order for the study to be even more accessible and interesting, some points of reflection are suggested (Note: in the absence of the number of lines, the name of the chapter, paragraph, any SUB, and the line number within the cited chapter, paragraph or sub are indicated: for example, "Abstract", paragraph "Methods", LINE 2 does not refer to line 2 of the entire document or page, but to line 2 of the "Methods" paragraph):

"Abstract", paragraph "Methods", LINE 2, in addition to the number indicated as recruited people "(n = 19,643)" it would be interesting to know, in principle, also the number of people who actually participated in the clinical study (n = 11,630) and the relationship that exists between these two numbers (for example: were they recruited but did not participate or was there abandonment after the initial recruitment?);

With reference to the same lemma, and to ensure greater accessibility in reading for a possible international reader, it would be of great help to use the official rule for punctuation in English, which provides for a comma to separate thousands and a period to separate decimal numbers. Just for example, the following cases are cited that could generate confusion:

"Abstract", paragraph "Methods", PAGE 2, LINE 2 "(n = 19,643)";

"Abstract", paragraph "Methods", PAGE 2, LINE 9 "(accessed by 4·5% of the sample)": for some people who refer to international numbering rules, it could be difficult to understand that it refers to a decimal number, which could be indicated by the notation "4.5%", same problem in "Abstract", "Findings", PAGE 2 LINE 2;

"PROCEDURES", PAGE 6, LINE 15: "Users participating in the trial were remunerated with ZAR 30 (approximately £1·50)": for some people who refer to international numbering rules, it could be difficult to understand that it refers to a decimal number, which could be indicated by the notation "£1.50%";

"RESULTS", LINE 1, "We recruited 19,643 women in 31 days. As expected, 20·0%": the same correction as the previous points is proposed for similar numbering;

"Abstract", paragraph "Methods" PAGE 2, LINE 9 "Primary outcomes were adapted (to?) validated scales of gender beliefs and past-month IPV. Secondary outcomes were identification of unhealthy relationship behaviors and depressive symptoms" would seem to be in contrast with some subsequent sentences in the rest of the document, such as: "We conducted a trial testing whether two versions of a behaviorally-informed chatbot could improve gender attitudes and skills in order to reduce IPV exposure among young women in South Africa." (INTRODUCTION, PAGE 3, LINE 37) or "Digital solutions represent an important approach to scale efforts to reduce IPV among young women. A behaviorally-informed chatbot helped women identify unhealthy relationship behaviors, improve gender equitable norms, and reduce exposure to IPV in the short-term." ("DISCUSSION", "Conclusion", PAGE 11, LINE 1) which instead would seem more consistent with the rest of the document. If the first hypothesis indicated in "Methods" is correct, perhaps it could be written in a way that leaves no doubt for the reader;

"Abstract", "Findings" paragraph, PAGE 2, LINE 1: a reader may be interested in, at this point, knowing which tool was used to evaluate "gender beliefs";

"Abstract", "Findings'' paragraph, PAGE 2, LINE 6: "Neither T1 nor T2 had a measurable effect on mental health" refers to a measurement of the absence of depression (PHQ-2). The extension of the concept of "absence of depression/anhedonia" generalized to the concept of "mental health," may lead the reader to believe that specific tools were used to evaluate "mental health," while instead, a brief screening tool was used to evaluate depression symptoms;

"INTRODUCTION", PAGE 3: the review section is treated very interestingly, a reader may be interested, in addition to the rightly cited information, if interventions on perpetrators/abusers have also been carried out in the same territories;

"INTRODUCTION", PAGE 3 LINE 5: "reported physical or sexual IPV" WHO defines IPV as "behaviour within an intimate relationship that causes physical, sexual or psychological harm, including acts of physical aggression, sexual coercion, psychological abuse and controlling behaviours." As can be seen from this definition, IPV already includes physical partner violence. It is understood that the citation in the text (4) reports exactly the wording of the results in table 2 (4), but probably a more effective wording can be chosen (also among those chosen in the research used as a source), to avoid that the term IPV is interpreted as exclusive for sexual violence and separate from physical violence, if both are practiced by the partner;

"MATERIALS & METHODS", SUB "Treatment Arms" SUB "Pure Control arm", PAGE 4 LINE 2 "The quiz pertaining to power equity was delivered directly after onboarding and eligibility questions.": from this sentence it seems that a questionnaire on "power equity" was administered. It would be interesting to read the questionnaire: is it a previously validated questionnaire or was it constructed ad hoc? What variables does it measure? Is it one of the tools mentioned in the "Measures" paragraph?

"MEASURES", PAGE 5, SUB "Gender equitable beliefs" LINE 3: What is the sample size in the validation assessment?

"MEASURES", PAGE 5, SUB "Mental Health" LINE 1: "Mental health was assessed using the brief screener of depressive symptoms, the two-item Patient Health Questionnaire (PHQ-2): as already noted in point 5, a reader may have incomplete information if it is not specified that the indices used do not measure "mental health" but the absence of depression/anhedonia;

Also in SUB "Socio-demographic," PAGE 5 LINE 5: "The state of users' mental health was assessed through one item asking if, in the past month, they had been mostly "energised and hopeful," "just getting along," or "down and hopeless." " can lead the reader to believe that tools for the evaluation of "mental health" were used;

"PROCEDURES", PAGE 6, LINE 6: "The trial was set up on RapidPro": an inexperienced reader may be interested in the reference to the software used and/or the organization that developed it;

“PROCEDURES”, PAGE 6, LINE 12: “To support 18,000 users on the chatbot…”: It could be indicated that the number is approximate or the precise number of people supported by the chatbot could be provided;

“RESULTS”, PAGE 8, LINE 6: “participants reported to have symptoms consistent with probable depression or anxiety…”: No instruments were indicated for evaluating anxiety;

“DISCUSSION”, PAGE 10, LINE 4 “The chatbot significantly improved skills around identifying unhealthy relationships, but did not alter mental health compared to the control.” and “DISCUSSION”, PAGE 10, LINE 35 “Prevalence of mental health symptoms among this relatively large sample of young women was high…”: As observed in points 5 and 10, the concept of "mental health" is broader than the absence of elements of depression/anhedonia, which makes it complex to refer to this construct without a reader wondering how it was measured;

“DISCUSSION”, “Limitations”: It is suggested to further reflect on the mode of data collection (online mode), which does not guarantee: 

that the person who initially declared themselves is actually the one who completed the courses;

that the person actually completes the activities;

it is also possible that the activity is promoted/maintained by the possible receipt of financial compensation;

Thank you for your valuable contribution to research, which touches on a very important and sensitive area for the health of all people. I hope these observations can be useful for a more specific reflection on some of the issues identified.

Best wishes for your work.

Reviewer #2: the Participants in the trial were 18 or older (24yr), what their education background or education level? In this study, the literacy level could play a crucial role. In this study, mental health was assessed using the brief screener of depressive symptoms in this study, there are more complicated to assess and identify depression.

6. PLOS authors have the option to publish the peer review history of their article (what does this mean?). If published, this will include your full peer review and any attached files.

**Do you want your identity to be public for this peer review?** For information about this choice, including consent withdrawal, please see our Privacy Policy.

Reviewer #1: Yes: Andrea Moi

Reviewer #2: No

---

## [Decision Letter · Decision Letter 1]

25 Aug 2023

Effects of digital chatbot on gender attitudes and exposure to intimate partner violence among young women in South Africa

PDIG-D-22-00354R1

Dear Dr Hatcher,

We are pleased to inform you that your manuscript 'Effects of digital chatbot on gender attitudes and exposure to intimate partner violence among young women in South Africa' has been provisionally accepted for publication in PLOS Digital Health.

Best regards,

Laura M. König

Academic Editor

PLOS Digital Health

Reviewer Comments (if any, and for reference):

Reviewer's Responses to Questions

**Comments to the Author**

1. If the authors have adequately addressed your comments raised in a previous round of review and you feel that this manuscript is now acceptable for publication, you may indicate that here to bypass the “Comments to the Author” section, enter your conflict of interest statement in the “Confidential to Editor” section, and submit your "Accept" recommendation.

Reviewer #1: All comments have been addressed

Reviewer #2: All comments have been addressed

2. Does this manuscript meet PLOS Digital Health’s publication criteria? Is the manuscript technically sound, and do the data support the conclusions? The manuscript must describe methodologically and ethically rigorous research with conclusions that are appropriately drawn based on the data presented.

Reviewer #1: Yes

Reviewer #2: Yes

3. Has the statistical analysis been performed appropriately and rigorously?

Reviewer #1: Yes

Reviewer #2: N/A

4. Have the authors made all data underlying the findings in their manuscript fully available (please refer to the Data Availability Statement at the start of the manuscript PDF file)?

Reviewer #1: Yes

Reviewer #2: Yes

5. Is the manuscript presented in an intelligible fashion and written in standard English?

Reviewer #1: Yes

Reviewer #2: Yes

6. Review Comments to the Author

Reviewer #1: We thank the editor and the authors of this interesting contribution and congratulate them for all the changes made that make the work very clear, understandable and interesting to the reader.

Reviewer #2: agree to acceptation

7. PLOS authors have the option to publish the peer review history of their article (what does this mean?). If published, this will include your full peer review and any attached files.

**Do you want your identity to be public for this peer review?** For information about this choice, including consent withdrawal, please see our Privacy Policy.

Reviewer #1: **Yes: **Andrea Moi

Reviewer #2: No
